# The Potential Role of Nutraceuticals as an Adjuvant in Breast Cancer Patients to Prevent Hair Loss Induced by Endocrine Therapy

**DOI:** 10.3390/nu12113537

**Published:** 2020-11-18

**Authors:** Giorgio Dell’Acqua, Aleksander Richards, M. Julie Thornton

**Affiliations:** 1Nutrafol, New York, NY 10016, USA; giorgio@nutrafol.com (G.D.); aleksander@nutrafol.com (A.R.); 2Centre for Skin Sciences, University of Bradford, Bradford BD17 7DF, UK

**Keywords:** breast cancer, estrogen receptor, hair follicle, aromatase inhibitors, tamoxifen, nutraceuticals, plant extract, endocrine therapy-induced hair loss

## Abstract

Nutraceuticals, natural dietary and botanical supplements offering health benefits, provide a basis for complementary and alternative medicine (CAM). Use of CAM by healthy individuals and patients with medical conditions is rapidly increasing. For the majority of breast cancer patients, treatment plans involve 5–10 yrs of endocrine therapy, but hair loss/thinning is a common side effect. Many women consider this significant, severely impacting on quality of life, even leading to non-compliance of therapy. Therefore, nutraceuticals that stimulate/maintain hair growth can be proposed. Although nutraceuticals are often available without prescription and taken at the discretion of patients, physicians can be reluctant to recommend them, even as adjuvants, since potential interactions with endocrine therapy have not been fully elucidated. It is, therefore, important to understand the modus operandi of ingredients to be confident that their use will not interfere/interact with therapy. The aim is to improve clinical/healthcare outcomes by combining specific nutraceuticals with conventional care whilst avoiding detrimental interactions. This review presents the current understanding of nutraceuticals beneficial to hair wellness and outcomes concerning efficacy/safety in breast cancer patients. We will focus on describing endocrine therapy and the role of estrogens in cancer and hair growth before evaluating the effects of natural ingredients on breast cancer and hair growth.

## 1. Estrogens and Breast Cancer

While estrogen has beneficial effects in many tissues, it also has a strong relationship with the initiation and development of endocrine-dependent cancers [1]. Breast cancer is the most common type of invasive cancer in women, accounting for over 500,000 deaths per year worldwide [2], and in the West, one in eight women will be diagnosed during her lifetime. Approximately 75–80% of all cases are estrogen receptor (ER)-positive [3]; therefore, increased estrogen exposure, e.g., early menarche, late menopause and long-term hormone-replacement therapy, is correlated with increased incidence [4]. This is further supported by data confirming that bilateral oophorectomy in women under 35 years of age reduces the lifetime risk by 75% [5]. Breast cancer is a heterogeneous disease, and transcriptomics have identified a number of molecular subtypes that are linked to diverse clinical outcomes. In addition to the ER, breast cancer cells can also express the progesterone receptor (PR) and/or HER2, an oncogene that belongs to human epidermal growth factor (EGF) receptor family. However, a proportion of breast cancers may be negative for all of these receptors (triple negative). The most common breast cancers are ER-positive and/or PR-positive and encompass two main molecular classifications: luminal A, which is HER2-negative and has low expression of Ki67, a marker of proliferation (35.6%), and luminal B, which may be either HER2-positive (13.1%) or -negative (22.5%) but has a high expression of Ki67 and grows faster [6]. Triple negative/basal-like breast cancer (15.2%) is more common in younger women, particularly those with *BRCA1* gene mutations, while the non-luminal HER2-positive subtype is ER- and PR- negative and, although not as common, tends to grow faster than luminal subtypes (13.7%) [6]. Interestingly, a recent study has also highlighted a protective role for parity and breastfeeding in breast cancer development, with parity associated with decreased risk of ER-positive breast cancer, while breastfeeding is inversely concomitant with hormone receptor-negative breast cancer [7].

Since 1996, it has been acknowledged that two distinct nuclear estrogen receptors (ERα and ERβ) exist, binding 17β-estradiol with comparable affinity [8,9]. The distribution and expression of ERα and ERβ is highly variable. ERα is the predominant receptor in female reproductive tissues, including the uterus and mammary glands, and also the principal receptor in pituitary, skeletal muscle, adipose tissue and bone. In contrast, ERβ is the principal receptor in the ovary, prostate, lung, cardiovascular system and central nervous system [10]. Activation of ERs increases transcriptional activity via interactions with palindrome estrogen response elements (ERE) located in the promoter region of estrogen-regulated target genes [11] (Figure 1). The recruitment of coactivator complexes and histone acetyltransferases, which are cell specific, are also necessary [12], with ERα and ERβ often having opposing requirements [13]. ERα and ERβ bind to EREs as homodimers or heterodimers; the cellular ratio of ERα/ERβ is thereby significant and impacts the response [14]. While classical estrogen signaling occurs via ERα/ERβ, many cells also express monomeric nuclear ERs that are trafficked from the cytoplasm following palmitoylation, via physical interaction with the caveolin-1 protein, transporting it to the caveolae rafts in the cell membrane [15,16]. Localization at the cell membrane works in partnership with cell membrane G-protein-coupled receptors (GPCRs), e.g., GPR30, to transmit rapid signals [17]. Although membrane-bound classical ERs lack the structural signaling domains of tyrosine kinase receptors, it appears that they can also transactivate the EGF or insulin-like growth factor 1 (IGF-1) receptors to stimulate kinase cascades [18]. 

While ERα is a key driver in ER-positive breast cancers, the molecular mechanisms of breast tumor initiation by estrogen are not fully understood. In normal mammary glands, ERα is tightly regulated via binding with 17β-estradiol, but this can become dysregulated, supporting tumor growth. It is thought that following conversion to quinone metabolites, estrogen binds directly to DNA, causing genotoxic effects and initiating mutations [19]. Activation of ERα by 17β-estradiol stimulates proliferation of these cells, which amass, forming a tumor [20]. As 75–80% of breast cancers are ERα-positive, therapies to inhibit ERα signaling are central to treatment. 

Currently, ERβ is not a diagnostic marker nor used in the treatment of breast cancer, and its role, if there is any, is unclear, although a number of studies suggest that it is an anti-oncogene [21]. In vitro studies indicate a role for ERβ in the inhibition of proliferation, migration and invasiveness of breast cancer cells [22,23]. Dysregulation of autophagy, which plays a key role in the maintenance of cellular homeostasis, has been implicated in many cancers. Recently, it has been suggested that an important regulatory role of ERβ in breast cancer is to induce autophagy [21,24]. Whether ERβ has a protective role in breast cancer is currently unknown, but understanding its mechanism of action may identify it as a potential target for new breast cancer treatments.

Another important feature in the development of breast cancer is the peripheral biosynthesis of estrogen from adrenal precursors (Figure 2). Breast tissue expresses high levels of aromatase, the enzyme required for the terminal biosynthesis of androgens into estrogens [25,26]. Furthermore, aromatase levels can vary in heterogeneous cells populations, with the contribution of aromatase in normal breast changing in magnitude with the development of a tumor [27] and, hence, increasing the local bioavailability of estrogen. 

## 2. Breast Cancer Treatment 

Until the 1970s, when new therapies began to emerge, aggressive mastectomy with removal of all glandular tissue including axillary lymph glands was the only approach available. In 1971, the non-steroidal triphenylethylene tamoxifen (brand name Nolvadex) went into a clinical trial and still remains the gold standard for the treatment of primary ER-positive breast cancer today [28,29]. While HER2-positive breast cancers grow faster than luminal cancers and can have a worse prognosis, patients with early-stage breast cancer are often treated successfully with trastuzumab (Herceptin), a HER2-specific monoclonal antibody targeted at the HER2 protein [30]. All breast cancer treatment encompasses radiotherapy, plus chemotherapy depending on the type of tumor, and while early-stage triple-negative breast cancer responds to chemotherapy, an optimal course of therapy still remains undefined [31]. 

Despite the success of tamoxifen as an ERα antagonist in breast cancer treatment, only 70% of ERα-positive patients respond, with 30% becoming resistant to endocrine therapy [32]. In addition, tamoxifen is correlated with an increased risk of endometrial cancer since it acts as an ERα agonist in the endometrium [33]. These observations that anti-estrogens can have mixed agonist/antagonist properties led to their redefinition as selective estrogen receptor modulators, or SERMs [34]. SERMs display mixed properties due to their individual cell-specific activity (Figure 1). While tamoxifen remains the gold standard for treatment of ERα-positive breast cancer in premenopausal women, aromatase inhibitors, including anastrozole, letrozole and exemestane, that suppress estrogen biosynthesis are the preferred option following menopause and in tamoxifen-resistant patients [35]. However, while tamoxifen only blocks ERα receptors in tissues where it acts as an antagonist (e.g., breast), due to tissue selectivity, it may have beneficial effects in other tissues where it acts as an agonist, e.g., bone. In contrast, aromatase inhibitors will block all estrogen action.

## 3. Estrogen and Hair Growth

The human hair follicle has an exceptional capacity for regeneration, cycling numerous times throughout adult life (Figure 3), and each cycle can result in a change in the hair fiber produced, e.g., in size or color [36]. When the growth phase (anagen) ends, the lower portion of the hair follicle undergoes programmed apoptosis and regression (catagen), followed by a resting, maintenance stage (telogen) before the existing hair is shed (exogen) and a new cycle commences. The time in each phase can alter over a lifetime, although on the scalp, approximately 80% of hair follicles are in anagen, which can last from 2 to 8 years [37]. During pregnancy, an increased number of hair follicles are maintained in anagen, resulting in denser hair. Postpartum, these additional anagen follicles enter telogen at the same time, causing temporary hair thinning [38]. Although estrogen levels fluctuate on a monthly basis and are high during pregnancy, menopause leads to a permanent state of hypoestrogenism due to programmed dissolution of ovarian estrogen biosynthesis [39]. Low menopausal estrogen levels correlate with a gradual diffuse thinning of scalp hair in women, along with changes in growth, diameter and pigmentation [40]. Notwithstanding, the link between hypoestrogenism and female pattern hair loss (FPHL) is still not well understood, although genome-wide association studies comparing aromatase (*CYP19A1*) and ERβ (*ESR2*) in almost 500 women affected with FPHL against 500 controls, demonstrated a nominal significant association with three single-nucleotide polymorphisms (SNPs) of each of the two genes [41,42].

A comparison of ERα and ERβ expression in human scalp has verified that ERβ is the principal ER in the hair follicle [43]. Cultured mesenchymal hair follicle cells (dermal papilla and dermal sheath), as well as interfollicular dermal fibroblasts, also predominately express ERβ [44]. Although menopause terminates ovarian function, peripheral estrogen biosynthesis from circulating adrenal androgens provides a source of estrogen post-menopause [45]. Similar to breast tissue, human hair follicles express the aromatase enzyme allowing them to regulate their own bioavailability of estrogens [46,47]. However, aromatase activity in human scalp varies with gender and anatomical region [48]. A recent study of women with FPHL has shown that they have a significantly lower expression of aromatase in their hair follicles, which will minimize estrogen bioavailability [49].

**Figure 3 nutrients-12-03537-f003:**
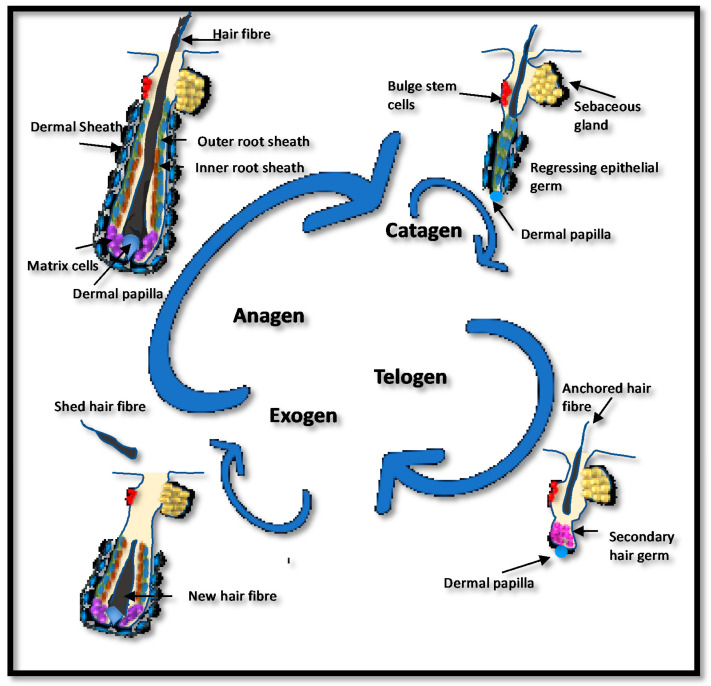
The human hair cycle. Hair follicles cycle throughout life with a growing phase (anagen) followed by regression (catagen) and maintenance (telogen). The hair fiber is shed during exogen, which usually coincides with the start of a new cycle. Bulb matrix cells proliferate and differentiate to produce the new follicle and hair fiber. In their center, the dermal papilla directs the type of hair produced. During catagen, the lower hair follicle regresses before entering telogen, where the hair fiber is firmly anchored but no further growth occurs. The length of anagen and telogen vary and will determine overall hair growth. If hair is shed before the initiation of a new anagen, the hair follicle may sit empty, a stage known as kenogen. Shortened anagen, lengthened telogen and increased exogen/kenogen can result in hair thinning. The number of hair follicles in kenogen is increased in women with female pattern hair loss (FPHL) [50].

## 4. Endocrine Therapy-Induced Hair Loss (ETIHL)

While the selective action of tamoxifen has beneficial effects on some estrogen-target tissues, e.g., bone, some women report scalp hair thinning as a side effect [51]. Likewise, data on aromatase inhibitors reported that up to 25% of the patients receiving treatment experienced hair loss or thinning [52]. Either blocking estrogen receptors with an antagonist, such as tamoxifen, or reducing estrogen bioavailability with aromatase inhibitors may inhibit the proliferation of scalp hair follicles [53] and induce them to enter the resting phase [54]. In a clinical study of 112 breast cancer patients on endocrine therapy, a patterned alopecia similar to androgenetic type (male pattern baldness) was confirmed by standardized clinical and trichoscopy images identifying the presence of vellus hairs and intermediate- and thick-diameter terminal hair shafts [55]. In 75 patients (67%) hair thinning was attributed to treatment with aromatase inhibitors and in 37 patients (33%) to treatment with tamoxifen. In 76% of cases, a more prominent recession of the frontotemporal area was observed. Treatment with topical minoxidil improved hair loss in 37 patients (80%), as documented through standardized clinical photographs of the scalp obtained at baseline and then after 3 or 6 months [55]. In a larger study of a cohort, 19,430 patients with endocrine-related cancers in 35 clinical trials were analyzed for effects on hair growth. Of these, 13,415 patients had received endocrine treatment, while 6015 patients served as controls. The incidence of all grades of alopecia ranged from 0% to 25%, with an overall incidence of 4.4% (95% confidence interval: 3.3–5.9%). The highest incidence was observed in patients treated with tamoxifen in a phase II trial (25.4%) [52]. Because ETIHL negatively affects sociocultural status and quality of life, it is a major problem and remains a therapeutic challenge in patients with breast cancer. Supplementation with vitamins (D, E, C, folic acid) and/or omega-3 fatty acids has been a popular adjuvant therapy, often associated in combination with minoxidil [35,56].

## 5. Estrogens as Antioxidants 

Although the role of intracellular ERs in estrogen signaling is well established, other ER-independent mechanisms have also been described [5]. Estrogen has been shown to have a protective effect against oxidative stress, which is independent of ERα and ERβ [57]. These antioxidant properties appear to be due to the structural properties of estrogen, in particular the phenolic A-ring that diminishes any reactive oxygen species (ROS) via a cyclic phenol-quinol mechanism [58]. The phenolic A ring structure of 17β-estradiol (Figure 4) can act as an antioxidant, providing antioxidant/redox cycling activity, which limits the release of ROS from damaged mitochondria, thereby protecting against cell damage [59]. These specific antioxidant properties can suppress oxidative stress induced by hydrogen peroxide, superoxide anions and other pro-oxidants by mechanism(s) independent of estrogen binding to the ERs [60]. Estrogen can protect against skin photoaging and improve the rate and quality of wound healing; however, these responses appear to be ER-dependent since both ERα and ERβ are expressed in human skin, including epidermal keratinocytes, dermal fibroblasts and hair follicle cells [43,44,46]. Furthermore, 17β-estradiol has been shown to exert a cytoprotective effect on cultured dermal fibroblasts derived from a patient with Friedreich’s ataxia, which are extremely sensitive to free radical damage and oxidative stress. Since these skin fibroblasts also express ERs, they were cultured in the presence of ICI 182,780, an antagonist of both ERα and ERβ and G15 the antagonist of the GPR30 cell membrane estrogen receptor, demonstrating that the cytoprotective effects of 17β-estradiol were independent of the ERs [57,59]. They concluded that these cytoprotective effects were dependent on antioxidant properties endowed by the phenolic structure of 17β-estradiol, since other phenolic compounds tested were protective, whereas all nonphenolic compounds were ineffective at reducing the levels of ROS.

## 6. Natural Ingredients and Their Effect on Breast Cancer and Hair Growth

Many plants can synthesize compounds that are structurally similar to 17β-estradiol. The crucial similarity that these phytoestrogens share with 17β-estradiol is their phenolic hydroxyl A ring, which plays a key role in docking to the ligand binding domains of ERα and ERβ, enabling them to mimic the effects of 17β-estradiol [61]. These hydrophobic polyphenolic compounds are considered to be naturally occurring SERMs since they can interact with both ERα and ERβ in either an agonistic or antagonistic manner [62]. Such nutraceuticals include resveratrol, a naturally occurring polyphenolic stilbene found in the skin of grapes, blueberries, raspberries and mulberries and in red wine [63]; tocotrienols, naturally occurring compounds found in plant seeds, such as rice bran, oil palm and annatto and belonging to the vitamin E family [64]; and brown seaweeds (kelp), which in addition to acting as an antagonist of ERα, are also inhibitors of aromatase, suggesting a protective role in the initiation and progression of estrogen-dependent cancers [65]. In contrast, saw palmetto (*Serona repens*) extract derived from the berries of the American dwarf tree is a competitive, nonselective inhibitor of both forms of 5α-reductase. It blocks nuclear uptake of 5α-DHT in target cells and decreases 5α-DHT binding to androgen receptors by approximately 50%. Additionally, the extract increases 3α-hydroxysteroid-dehydrogenase activity, increasing the conversion of 5α-DHT to its weaker metabolite, androstanediol [66]. 

Many nutraceuticals also exhibit strong anti-inflammatory and antioxidant properties. For example, astaxanthin, a carotenoid produced when the freshwater algae *Haematococcus pluvialis* is subjected to stress, it is a powerful antioxidant superior to other carotenoids [67]. It preserves the integrity of the cell membrane via its insertion into the lipid bilayer, thereby protecting the redox state and integrity of the mitochondria by decreasing ROS and stimulating the production of antioxidants, such as superoxide dismutase (SOD), catalase and glutathione (GSH) [68,69]. Maca (*Lepidium meyenii*) extract is derived from the roots of the plant found in the Andean region of South America and contains high levels of flavonolignans and glucosinolates that have been reported to have anti-proliferative activity against several cancers, including breast cancer [70,71]. The health benefits of curcumin, the main secondary metabolites of turmeric (*Curcuma longa*) extract, which is derived from the roots of the plant mainly grown in India, have been linked to its anti-inflammatory and antioxidant action [72]. Horsetail (*Equisetum arvense)* another plant extract used in traditional medicine also exhibits strong anti-inflammatory and antioxidant properties [73]. A plant extract which is particularly rich in steroid withanolides that exhibits anti-inflammatory properties is ashwagandha (*Withania somnifera*) [74]. It has well-documented anti-cancer properties and has been shown to suppress ER-α protein level by about 90% in breast cancer cells, while increasing the expression of ER-β protein by about 20–30% [75]. Nutritional factors appear to play a role in persistent increased hair shedding—e.g., serum ferritin concentrations factor in female hair loss [76] and supplementation with omegas 3 and 6 can reduce hair loss by improving hair density and reducing the percentage of telogen hair follicles [77]. Recently, it has been demonstrated that nutraceuticals containing Annurca apple polyphenols (*Malus pumila Miller* cv. Annurca) can promote human hair growth, increasing hair density, weight and keratin content both in vitro and in vivo [78]. A summary of the protective and inhibitory effects of different nutraceuticals on breast cancer and their potential effect on improving hair growth is given in Table 1.

## 7. Discussion

This review has highlighted an important role for sex hormones in the development of FPHL and that endocrine therapy in the form of tamoxifen and/or aromatase inhibitors for the management of breast cancer can induce ETIHL. For some women, this is a considerable side effect which can significantly impact the quality of life and, in some cases, non-compliance of necessary endocrine treatment. For many women, breast cancer treatment may also include chemotherapy, which causes a dramatic increase in the level of free radicals and reactive oxygen species in the body. Since 17β-estradiol can also act as an antioxidant, its eradication with aromatase inhibitors can further exacerbate the problem.

Common intervention treatments are supplements containing vitamins and omega fatty acids; however, botanicals have a much higher therapeutic potential, although with medical professionals having some concerns around ER activity and activation. While some physicians have concerns around the use of plant-based nutraceuticals to treat hair loss in women that are undergoing endocrine therapy for the treatment of breast cancer, there is evidence to show that nutraceuticals such as curcumin, ashwagandha, maca, Annurca apple fruits, safflower and ginseng do not stimulate the proliferation of breast cancer cells. This may be due to both their tissue selectivity (SERMs with antagonistic properties in breast cancer cells) or their higher affinity for ERβ. If these compounds have SERM activity, the ideal scenario is that their activity is as an antagonist in breast cancer cells and is an agonist in the hair follicle. Since the predominant ER in the human hair follicle is ERβ and in breast cancer cells it is ERα, then their relative affinity for these two distinct receptors is also of importance. Furthermore, there is evidence to show that nutraceuticals such as curcumin, tocotrienols, kelp, ashwagandha and resveratrol, in addition to exerting an anti-proliferative effect on breast cancer cells, can also downregulate ERα while stabilizing the anti-proliferative ERβ, thereby altering the ERα:ERβ ratio. In addition, *Capsicum annuum* extracted from chilies can also reduce the expression of HER2 in breast cancer cells. Therefore, the potential use of any of these nutraceuticals to promote hair growth in women should not pose an increased risk of breast cancer. 

However, more extensive studies are required to understand how these nutraceuticals can modulate ER expression in breast cancer cells, or whether they actually have no interaction with the ERs and purely have anti-inflammatory and/or antioxidant actions. Indeed, all of them exhibit anti-inflammatory and antioxidant activity and it may be that since many of them are polyphenols, this is their mode of action. The epidermal matrix cells of the anagen hair follicle are the second highest proliferative cells in the human body, which is why hair is so drastically impacted by chemotherapy. Consequently, the hair follicle is susceptible to oxidative stress and the antioxidant mechanism action of these nutraceuticals may be the main driving force behind their stimulatory/protective influence.

The most important aspect of ingredient safety in supplements that can improve ETIHL for women undergoing breast cancer treatment is that the ingredients should not be acting as estrogen that stimulates the proliferation of breast cancer cells in postmenopausal women, or that they compete with the antagonistic action of tamoxifen. Therefore, understanding their mechanism of action is key, since many women will be prescribed endocrine-directed therapy in the form of tamoxifen, aromatase inhibitors or Herceptin. Although the exact mechanisms are unclear, this literature review highlights evidence advocating that supplements containing resveratrol, saw palmetto, maca, curcumin, tocotrienols, ashwagandha, horsetail, astaxanthin, kelp, capsicum, Annurca apple fruits, safflower and ginseng can have beneficial effects on hair growth, without adverse effects on breast cancer patients. 

## Figures and Tables

**Figure 1 nutrients-12-03537-f001:**
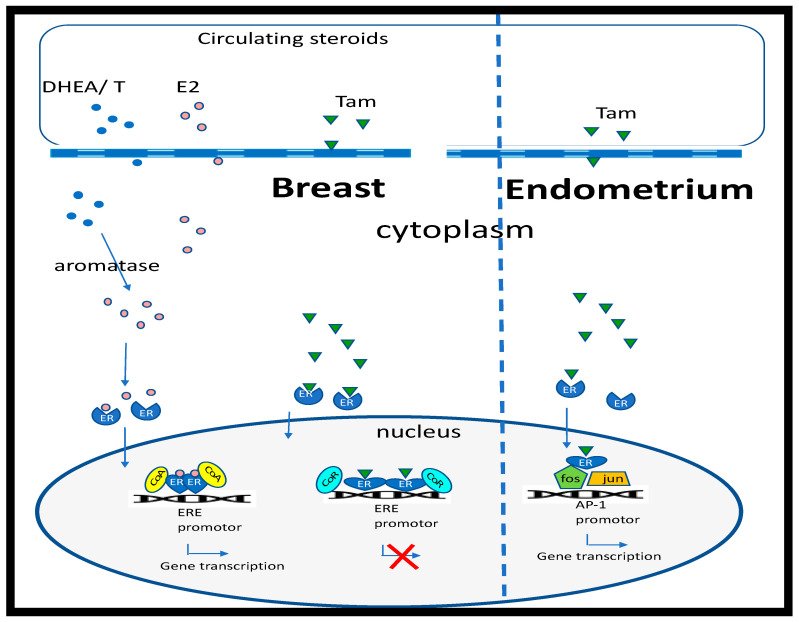
Regulation of estrogen-response genes by 17β-estradiol and tamoxifen. Estradiol (E2) (pink circles) passes through the cell membrane and binds to the estrogen receptor (ER), inducing a conformational change in shape and nuclear translocation, where it interacts with specific estrogen response elements (ERE) that regulate estrogen responsive genes, after recruiting cell-specific cofactors (CoA). E2 can be metabolized by aromatase from androgen precursors (blue circles; dehydroepiandrosterone (DHEA) or testosterone (T)). In breast cells, binding of tamoxifen (Tam) to ERs results in a conformational change that recruits corepressors (CoR) of gene transcription. However, in the endometrium, tamoxifen binding to the ER results in protein:protein interactions and activation of the activator protein 1 (AP-1) promotor.

**Figure 2 nutrients-12-03537-f002:**
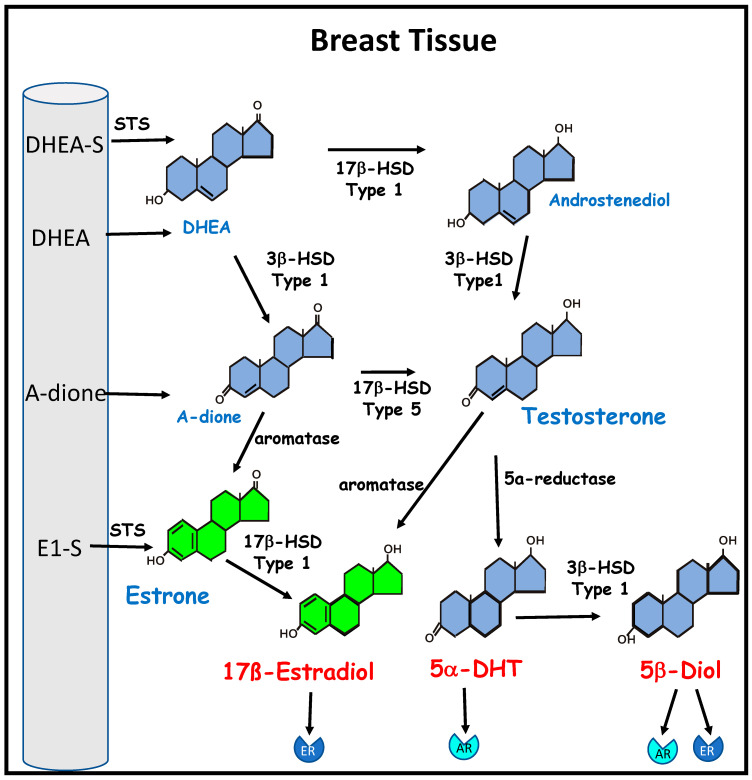
The biosynthesis of estrogen from inactive circulating precursors in breast cancer. Bioactive estrogen can be synthesized by breast cancer cells and breast stromal tissue from circulating precursor steroids. These include the adrenal androgens dehydroepiandrosterone (DHEA) and androstenedione (A-dione); dehydroepiandrosterone sulphate (DHEA-S), the major circulating androgen, and estrone sulphate (E1-S), the major circulating estrogen in postmenopausal women, which can be converted to DHEA and estrone (E1), respectively, by steroid sulfatase (STS). DHEA and androstenediol are metabolized by 3β- hydroxysteroid dehydrogenase (3β-HSD) type 1 to the estrogen precursors A-dione and testosterone. Testosterone can be metabolized by 5α-reductase to the potent androgen 5α-dihydrotestosterone (5α-DHT), which has a high affinity for the androgen receptor (AR). Type 1 3β-HSD can convert 5α-DHT to 5α-androstane-3β,17β-diol (5β-diol), which is an agonist of both the AR and the estrogen receptor (ER). Aromatase is required for the conversion of testosterone to 17β-estradiol and conversion of A-dione to E1, which can be further metabolized by 17β- hydroxysteroid dehydrogenase (17β-HSD) type 1 to 17β-estradiol, which has a high affinity for ER.

**Figure 4 nutrients-12-03537-f004:**
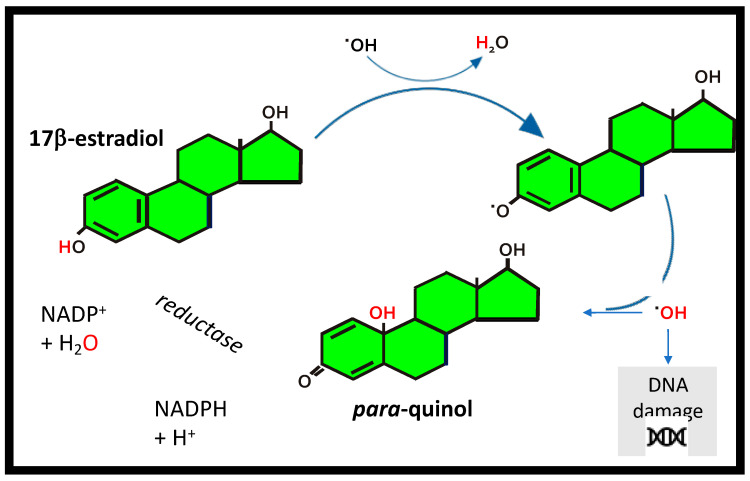
17β-estradiol as an antioxidant. The antioxidant cycle for 17β-estradiol by transfer of a H-atom to a free radical (^•^OH), to form a phenoxyl radical that scavenges ^•^OH forming a para-quinol which undergoes enzymatic reduction in the presence of the cofactor nicotinamide adenine dinucleotide phosphate (NADPH) to convert back to 17β–estradiol.

**Table 1 nutrients-12-03537-t001:** A summary of the estrogenic and antioxidant properties of nutraceuticals that provide a basis for complementary and alternative medicine in both breast cancer and hair growth. Their protective and inhibitory effects on breast cancer are summarized in addition to a potential role for the improvement of hair growth.

NUTRACEUTICAL	PROPERTIES	BREAST CANCER	HAIR GROWTH	POTENTIAL ADJUVANT
Resveratrol	Naturally occurring polyphenolic stilbene found in, blueberries, raspberries, mulberries, grapes and red wine. Can signal via ERα and ERβ. An effective antioxidant with strong anti-inflammatory properties.	Inhibits estrogen-induced breast carcinogenesis via induction of NRF2-mediated protective pathways [79]. Induction of apoptosis in ER +ve MCF-7 cells is via inhibition of the ERα-dependent PI3K pathway [80]. Resveratrol analogues combined with tamoxifen have a synergistic effect on inhibiting proliferation of ER +ve and ER-ve breast cancer cells [81].	Using the sensitive probe DCFH-DA, it was shown to significantly reduce oxygen peroxide-induced oxidative stress generated in hair follicles and hair matrix cells [82]. Furthermore, a clinical study of 79 women suffering from hair loss treated with a topical combination of pyridine-2, 4-dicarboxylic acid diethyl ester and resveratrol reported significantly increased hair density after 1.5 months [82].	Anti-carcinogenic. Synergizes with tamoxifen. Anti-inflammatory. Antioxidant. Potential to improve hair density in women and reduce oxidative stress.
Tocotrienols	Hydrophobic phenolic antioxidants with structural features allow binding to ERα and ERβ. Found in plant seeds, e.g., rice bran, oil palm and annatto; belong to the vitamin E family [64].	Exhibit high affinity for ERβ and promote nuclear translocation, modulating cell morphology, caspase-3 activation, DNA fragmentation and apoptosis [83,84]. Binding to ERβ induces apoptosis in breast cancer cells [84,85]. Can synergize with tamoxifen to inhibit proliferation [86] with a clinical trial suggesting breast cancer survival may be extended by combining tocotrienol with tamoxifen therapy [87]. Proliferation of ER-negative breast cancer cell line can also be inhibited by α-, γ- and δ-tocotrienols [86,88].	Induce murine hair follicle development and stimulate anagen hair cycling by suppressing epidermal E-cadherin followed by a 4-fold induction of β-catenin and its nuclear translocation [89]. In an 8-month treatment of 38 patients with hair loss, increased hair counts in 34.5% [90]; thought to be due to inhibition of lipid peroxidation and reduction in oxidative stress in the hair follicle.	ERβ agonist. Anti-carcinogenic. Synergizes with tamoxifen. Antioxidant. Potential to improve hair cycling and reduce oxidative stress.
Saw palmetto (*Serenoa repens*)	American dwarf tree berries. Competitive, 5α reductase (1 and 2) inhibitor. Multiple sites of action—different pharmacodynamic profile to finasteride [66]. Increases 3α-hydroxysteroid-dehydrogenase activity, converting 5α-DHT to a weaker metabolite, androstanediol [66].	Induces dose-dependent inhibition of ER + ve/−ve breast cancer cell line proliferation. Inhibition 2.5x greater (*p* < 0.01) in MCF-7 cells (ER +ve) than MDA MB231 cells (ER −ve) [91]. The anti-proliferative effect was triggered by the induction of apoptosis. Has anti-estrogenic activity in prostate tissue of men with BPH [92]. Furthermore, decreases 5α-DHT and estradiol plasma levels in men [93].	Most studies have been conducted on men with androgenetic alopecia. Daily treatment with 200 mg in 26 men with androgenetic alopecia saw improvement in 60% compared to 11% with placebo [94]. A daily oral supplement of 320 mg compared to 1 mg finasteride in 100 men saw more growth on frontal and vertex scalp in 68% of the finasteride cohort, while 38% of the saw palmetto group had increased hair growth on the vertex [95]. A topical application saw increased terminal hair counts after 12 and 24 weeks [96].	5α-reductase inhibitor. Anti-carcinogenic. Potential to improve terminal hair counts, particularly on vertex.
Maca (*Lepidium meyenii*)	Plant roots (Andes) contain high levels of flavonolignans and glucosinolates. Protective against inflammation, increases antioxidants and has hormonal balancing properties. Supports the antioxidant system by activating SOD and GSH [97].	Has anti-proliferative activity against cancer, including breast cancer [63,64]. In rats with BPH), oral supplementation of a butanol fraction of red maca reduces prostate size by restoring ERβ expression without changing ARs and ERα expression [98]. In postmenopausal women, after 2 months, Maca stimulated estradiol and suppressed cortisol [99]. It modulates estrogen levels by targeting only ERβ and/or re-establishing hormonal homeostasis through the hypothalamus–pituitary–ovarian axis [99].	IL-6 has been implicated in forms of hair loss such as AA [100]. Blood analysis of individuals consuming Maca has shown a reduced level of IL-6 [101], suggesting potential application for treatment of AA. Maca has the ability to support the antioxidant defense system by activating SOD and GSH [97], which are powerful antioxidants that lower levels of oxidative stress in forms of hair loss such as AGA [102].	Anti-carcinogenic. Anti-inflammatory. Antioxidant. Potential to reduce oxidative stress in the hair follicle and improve hair growth.
Curcumin	Curcumin is the main metabolite in turmeric (*Curcuma longa*) root, with anti-inflammatory and antioxidant properties [72]. It down-regulates inducible nitric oxide and cyclooxygenase-2; inhibits nuclear factor-kB signaling, decreasing pro-inflammatory cytokines, e.g., TNF-α and IL-1, which are implicated in hair loss [103]. It is recognized as a strong anti-cancer agent, including breast cancer, in traditional medicine [104].	Inhibits proliferation of T-47D breast cancer cells in a dose-dependent manner by downregulating ERα [105,106]. It has antioxidant and anti-inflammatory action and can induce cell cycle arrest in breast cancer cells [107]. Studies on other estrogen-sensitive cancer cell lines, have demonstrated a capacity to counteract the proliferative effect of estradiol [108] as well as the synthesis of estradiol [109].	Encapsulating curcumin in liposomes enhances penetration in porcine hair follicles by 70% [110]. Similarly, a cyclodextrin complex enhances curcumin follicle penetration [111]. *In vitro* it blocked expression of genes that inhibit hair growth, e.g., PAK1 and TGF-β1 [112,113]. Topical application (5%) or in combination with 5% minoxidil was assessed in 87 men with androgenetic alopecia for 6 months, demonstrating that while curcumin alone did not stimulate hair growth, a combination with minoxidil showed a significant improvement, suggesting they may act in a synergistic manner [114].	Anti-carcinogenic. Anti-inflammatory. Antioxidant. Synergizes with minoxidil with the potential to improve hair growth.
Ashwagandha (*Withania somnifera*)	Derived from the root and leaf of the plant, which is particularly rich in withanolides, which are the main active ingredients and have anti-inflammatory and adaptogenic properties [74]. Ashwagandha and its actives have been proposed as anti-cancer agents that are able to modulate apoptotic, proliferative and metastatic markers in cancer [115,116].	In breast cancer, it demonstrates chemo-preventive activity in female rats following administration of the mammary carcinogen methylnitrosourea by significantly reducing the rate of cell proliferation in mammary tumors [117]. In MCF-7 cells, withaferin, the main active ingredient, suppressed ER-α protein by 90%, while expression of ER-β protein increased by 20–30% [75]. Withaferin-mediated down-regulation of ER-α protein expression correlated with a decrease in its nuclear level, suppression of its mRNA level, and inhibition of estrogen-dependent activation of ERE2e1b-luciferase reporter gene [75]. In another study, it was shown that ashwagandha inhibited proliferation and induced apoptosis in the MCF-7 breast cancer cell line by down-regulating ERα protein levels via proteasome-dependent ERα degradation [118]. Furthermore, withaferin alters the mitochondria dynamic resulting in apoptosis in breast cancer cells [119] and impairs cancer autophagy resulting in proteotoxicity and death [120]. Since withaferin has shown activity in triple negative breast cancer cells MDA-MB-231, by blocking their invasiveness and associated markers, the active has been suggested for further (pre)clinical development to defeat aggressive metastatic breast cancer [121].	Ashwagandha extract has anti-stress and anti-anxiety properties [122]. Being a powerful adaptogen, the extract has the ability to keep cortisol levels at a healthy homeostatic level while improving one’s resistance to stress [123,124]. Chronic stress activates the HPA axis by increasing cortisol levels, which, in turn, inhibits the HPT axis and reduces serum T3 and T4 levels [125]. Endocrine disorders, such as hypothyroidism, can cause hair loss [126]. Treatment with ashwagandha lowers serum cortisol levels by downregulation of the HPA axis, which, in turn, upregulates the HPT axis to normalize the thyroid hormone level [127]. Thyroid hormones, such as T3, are integral to hair growth and wellness [128]. High-concentration, full-spectrum ashwagandha root extract can improve resistance towards stress and anxiety [129]. Anxiety can contribute to hair loss as evidenced by a study finding a high prevalence of anxiety symptoms in patients with alopecia areata [130]. Stress has been associated with hair loss, such as in the case of telogen effluvium, characterized by a non-scarring, non-inflammatory alopecia of relatively sudden onset caused by physiologic or emotional stress [131]. Use of ashwagandha would then reduce the impact of stress on hair loss.	Anti-inflammatory. Anti-carcinogenic. Suppresses ERα. Increases ERβ. Reduces cortisol. Normalizes thyroid hormone levels. No direct hair growth studies, but effect on circulating cortisol and T3 will impact hair growth.
Horsetail (*Equisetum arvense)*	Used in traditional medicine and has strong anti-inflammatory and antioxidant properties [73]. Down-regulates TNF-α and upregulates anti-inflammatory IL-10 in rheumatoid arthritis patients [132]. Suppresses free radicals and inflammatory mediators in IFN- γ and LPS-stimulated murine macrophages [133].	Extracts demonstrate antioxidative effects in two lipid peroxidation systems and anti-proliferative activity in human tumor cell lines [134].	Extracts inhibit 5α-reductase and decrease IL-6 secretion in LPS-stimulated macrophages and are not toxic against human follicle dermal papilla cells [135].	Anticarcinogenic. Anti-inflammatory. Antioxidant. 5α-reductase inhibitor.
Astaxanthin	A carotenoid produced by *Haematococcus pluvialis* (fresh-water algae). This blood-red pigment accumulates when algae are subjected to stress [67]. A powerful antioxidant, superior to other carotenoids. It preserves membrane integrity by insertion in the bilayer, protecting the redox state and mitochondrial function. It decreases ROS and increases production of antioxidants, e.g., catalase, GSH and SOD [68,69].	In ER +ve MCF-7 cells, it causes a significant accumulation of cells in the G2/M phase [136]. It significantly reduces proliferation and migration of MCF-7 cells [137] and, after 24 hours, leads to a significant decrease in viability [138]. It also significantly reduces T47D viability in a dose-dependent manner [139]. Its ability to reduce proliferation and viability and to increase apoptosis of cancer cells is associated with an increased expression of PPARγ [140]. Another mechanism by which it inhibits breast cancer cells by reducing proliferation is via inactivation of the PI3K/AKT pathway [140].	Dysfunction of mitochondrial respiration delays hair regeneration [141]. Astaxanthin can protect mitochondria from oxidative damage by helping the scavenging of ROS from follicle cells [68].	Powerful antioxidant. Anti-carcinogenic. Has potential to improve hair growth by protecting from oxidative stress.
Kelp (brown seaweed)	Brown seaweeds rich in polysaccharides (e.g., alginic acid and fucoidan), vitamin B12, iron, iodine, phlorotannins and fucoxanthin, with biological properties [142]. In rats, it has a dual action as an antagonistic of ERα and as an aromatase inhibitor, suggesting a protective role in the initiation and progression of estrogen-dependent cancers [65].	Treatment of MCF-7 and MDA-MB-231 breast cancer cells with *Sargassum muticum* methanol extract significantly inhibits proliferation and angiogenesis and increases apoptosis in a time- and dose-dependent manner [65,143]. Polysaccharides from *Sargassum wightii* induce apoptosis of MCF7 and MDA-MB-231 cells by increasing ROS, mitochondrial membrane cleavage and nuclei damage [144]. In animal models of breast cancer, fucoidan, one of the main polysaccharides in seaweed, stimulated an immune response [145]. As an adjuvant it increased sensitization to radiation and chemotherapy in breast cancer cells and animal models [146,147,148]. The co-administration of fucoidan on pharmacokinetics of letrozole and tamoxifen in patients with breast cancer showed it was well tolerated and did not alter plasma concentration of the drugs [149].	An extract from brown seaweed *Undariopsis peterseniana*, rich in fucoxanthinone, promoted hair growth in rat hair follicles *ex vivo* and stimulated rat dermal papilla cell proliferation by activating the Wnt/β-Catenin pathway [150]. The brown alga *Ecklonia cava*, rich in phlorotannins, stimulated human follicle dermal papilla cell proliferation and hair fiber elongation in human hair follicles *ex vivo* and hair growth in mice [151,152]. Fucoidan, a prominent polysaccharide in brown seaweed, stimulated production of HGF, which stimulates the hair cycle [153,154]. A patent on fucoidan as a hair-restoring agent was filed in 2000 (EP1234568B1). A double-blinded, placebo-controlled clinical trial showed seaweed extracts in supplements helped to prevent patterned hair loss and promoted scalp health in men and women [155]. A study on mice showed a mixture of seaweed extracts was as effective for hair growth promotion as minoxodil [156].	Anti-carcinogenic. Does not interfere with plasma levels of tamoxifen or aromatase inhibitors. Contains supplements important for hair growth, e.g., B12 and iron [77]. Direct effects on dermal papilla cells and hair growth.
*Malus pumila Miller* cv. Annurca (apple fruits)	Polyphenols with a high content of oligomeric procyanidins, specifically procyanidin B2.	Apple polyphenolic compounds had a significant antiproliferative action on MCF-7 cells. An amount of 500 μM of Annurca flesh polyphenols extract (AFPE) induced a cell cycle arrest at G2/M. AFPE was also capable of inducing morphological changes as evidenced by nuclear condensation [157].	Significantly stimulates the synthesis of cytokeratins in the human keratinocyte HaCaT cell line [78] and primary cultures of human hair keratinocytes [158] in vitro and protects cultured human dermal papilla from oxidative stress [158]. Oral supplementation (800 mg) significantly increased hair growth, density and keratin content in a cohort of 250 patients (116 men and 134 women; 30–83 years of age) randomly divided into two subgroups (each one of 125 subjects, 58 men and 67 women) after 2 months [78].	Strong antioxidant. Anti-carcinogenic. Stimulates keratin synthesis. Stimulates hair growth and density.
*Carthamus Tinctorius* L. (Safflower)	Active constituents include flavonoids, phenylethanoid glycosides, coumarins, fatty acids and steroids. Its oil has high nutritional value, consisting of 70% polyunsaturated fatty acid (i.e., linoleic acid) and 10% monounsaturated oleic acid.	Significantly increases apoptotic rate of the MCF-7 cells in a dose-dependent manner by down-regulating expression of Bcl-2 and upregulating Bcl-2-associated X protein in a time-dependent manner. Additionally, it significantly reduced expression of MMP-9 increased expression of TIMP-1 [159].	Suppresses the expression of TGF-β1 and significantly increases length of hair follicles in culture by stimulating the expression of VEGF and KGF [160]. Promotes hair growth, at least in part, by upregulating expression of β-catenin [161].	Antioxidant. Anti-inflammatory. Anti-carcinogenic. Stimulates hair growth.
*Capsicum annuum*	Chile pepper with high phenolic content and antioxidant activity.	Capsaicin is potent inhibiter of ER +ve (MCF-7, T47D, BT-474) and ER-ve (SKBR-3, MDA-MB231) breast cancer cell lines, associated with G0/G1 cell-cycle arrest, increased levels of apoptosis and reduced protein expression of human epidermal growth factor receptor (EGFR), HER2, activated extracellular-regulated kinase (ERK) and cyclin D1. Further blocked breast cancer cell migration in vitro and decreased tumors by 50%, growing orthotopically in immunodeficient mice [162].	Significantly increases IGF-I production in har follicles, promoting hair growth [163]. Capsaicin as an isolated active, when injected intradermally into the back skin of C57BL/6 mice with all follicles in the telogen phase of hair cycle, induced significant hair growth (anagen), which was associated with substantial mast cell degranulation [164].	
*Panax ginseng*	The three key ingredients are saponins (ginsenoside), polysaccharides and phenolic compounds. Ginsenosides are categorized into two groups based on their chemical structure, i.e., oleanane type (five-ring structure) and dammarane type (four-ring structure) [165].	Anticancer properties include induction of apoptosis, blocking angiogenesis, and inhibiting proliferation in cancer cell lines including MCF-7. [166]. Inhibits breast cancer cell proliferation and both anchorage-dependent and -independent breast cancer cell colony formation. In addition, it decreased the stability of the IGF-1R protein in breast cancer cells. suggesting that IGF-1R is an important target for treatment and prevention of breast cancer [167].	In clinical studies, red ginseng combined with topical minoxidil increases its effectiveness at promoting hair growth in human clinical studies. Moreover, it promotes the proliferation of human dermal follicle papilla cells and keratinocytes and enhances hair anagen in the mouse [165]. In chemotherapy-induced alopecia, it can protect against premature catagen [168], and *in vitro*, it has been shown to stimulate proliferation and inhibit apoptosis in hair follicle outer root sheath keratinocytes [169].	Anti-carcinogenic. Antioxidant. Anti-inflammatory.

**Abbreviations:** AA = alopecia areata; AFPE = Annurca flesh polyphenols extract; AGA = androgenetic alopecia; AR = androgen receptor; Bcl-2 = B-cell lymphoma 2; BPH = benign prostatic hyperplasia; DCFH-DA = dichloro-dihydro-fluorescein diacetate; EGFR = epidermal growth factor receptor; ERK = extracellular signal-related kinase; GSH = glutathione; HGF = hepatocyte growth factor; HPA = hypothalamic-pituitary-adrenal axis; HPT = hypothalamic-pituitary-thyroid axis; IFN = interferon; LPS = lipopolysaccharide; IGF-1R = insulin-like growth factor 1 receptor; IL = interleukin; KGF = keratinocyte growth factor; MMP-9 = matrix metalloproteinase-9; NRF2 = nuclear factor erythroid 2-related factor 2; P13K = phosphatidylinositol 3-kinase; PAK1/AKT = serine/threonine protein kinase/protein kinase B; PPARγ = peroxisome proliferator-activated receptor gamma; ROS = reactive oxygen species; SOD = superoxide dismutase; T3 = triiodothyronine; TGF-β = transforming growth factor beta; TIMP-1 = tissue inhibitor of metalloproteinase-1; TNF-α = tumour necrosis factor alpha; VEGF = vascular endothelial growth factor; Wnt = wingless-related integration site.

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
