# Peer review of "The Potential Role of Nutraceuticals as an Adjuvant in Breast Cancer Patients to Prevent Hair Loss Induced by Endocrine Therapy"

_nutrients, 2020, doi:10.3390/nu12113537_

Round 1

Reviewer 1 Report

It is a nice review, well written, and presenting the subject of chemotherapy-induced hair loss from a personal and peculiar point of view.

However, the review is not complete and misses many data produced by several groups in the last years and describing the effect exerted by several nutraceuticals on chemotherapy-induced alopecia ( just to mention some of them: Carthamus tinctorius, Malus Pumila, Prunus persica, Zingiber officinalis, Panax ginseng, Salvia officinalis, Cuscuta epithimum, Carum petroselinum, Angelica archangelica, Capsicum annuum)

Here a list of references to be included and discussed in the review or simply to be inspired by for further searches:

Famenini S (2014) J Drugs Dermatol 13: 809-812.

Riccio G (2018) Nutrients Nov 20;10(11):1808.

Rushton DH (2002) Clin Exp Dermatol 27: 396-404.

Le Floc'h C, (2015) J Cosmet Dermatol 14: 76-82.

Lengg N, (2007) Therapy 4: 59-65.

Candiani C, Bestetti A (2010). European Bulletin of Drug Research 18: 9-17.

Gori G, (2010) European Bulletin of Drug Research 18: 1-4.

Author Response

Attachment

Reviewer 2 Report

The work is well organized and easily understandable.

The authors may improve and deepen the section about the different breast cancer classification (section 1) and the Breast Cancer Treatment (section 2).

Among nutraceuticals, they could also argue about the effects of polyphenols, as strong antioxidants. Moreover, it is also known in literature that polyphenols are important for the synthesis of hair keratins and therefore for hair growth (Nutrients, 2019, 11(12), 3041; J Med Food, 2018, 21(1):90-103.).

At last, the authors could improve the quality of the images. In particular, figure 3 is blurred.

Author Response

Attachment

Reviewer 3 Report

This review describes the potential role of neutraceutical as an adjuvant in breast cancer patients. The authors describe in detail the pathomechanisms and treatment of breast cancer. Furthermore, the influence of estrogens on hair growth and the influence of endocrine therapy on hair loss are described. In a final section natural ingredients and their effect on breast cancer are described.

All in all, the pathomechanism as well as the treatment of breast cancer is described here in great detail.

However, the actual title of the review regarding the influence of nutraceuticals is not sufficiently described. A number of publications are listed in a table. But it would be nice if they were described in more detail and based on this, the authors would give an outlook.

  1. Section: Estrogen and Breast cancer, Line 29: Please insert a section on breast feeding and reduced risk of breast cancer
  2. Section: Estrogen and hair, Line 125: Normally up to 80% are in the anagen phase. Please change.

Line 126: Please indicate the period from when to when a hair can be in the growth phase. For not all people it is 8 years. The "hair length" varies from person to person.

Line 127: To avoid misunderstandings for readers: Please do not write thicker hair but denser hair.

Line 133: please add here that FPHLis linked to polymorphisms of the ERß gene.

Yip L, Zaloumis S, Irwin D, Severi G, Hopper J, Giles G, et al. Association analysis of oestrogen receptor beta gene (ESR2) polymorphisms with female pattern hair loss. Br J Dermatol 2012; 166:1131 - 4

Yip L, Zaloumis S, Irwin D, Severi G, Hopper J, Giles G, et al. Gene-wide association study between the aromatase gene (CYP19A1) and female pattern hair loss. Br J Dermatol 2012; 161:289 – 94

Line 154: Please add that the number of kenogen phases increased in female AGA

Line 161-164: Was a statement made here about the anagen rate?

  1. Section Estrogens as Antioxidants: With regard to the effect of estrogen as an antioxidant, two or three more sentences could be described with regard to its influence on the skin (UV stress, wound healing etc.)
  2. Section: Natural Ingredients and their effects on Breats Cancer and hair growth:

This section is actually the most important part of the review and should be completely revised again. On the one hand, the phytoestrogens should be described here once again and their positive/negative effects should be worked out. Therapies and their mechanism of action in the case of hair loss should also be briefly mentioned in order to then weigh up to what extent they make sense or not in the case of breast cancer. Overall, the publications listed in the table should be explained in more detail.

Author Response

Attachment

Round 2

Reviewer 1 Report

The manuscript can be accepted for pubblication.

Reviewer 3 Report

Good work! The article is much improved. The corrections previously required were performed.  In addition, the last section is improved after editing the table and the written part.

Thus, it is possible to accept the article without further corrections .